# HLA-DPB1 Reactive T Cell Receptors for Adoptive Immunotherapy in Allogeneic Stem Cell Transplantation

**DOI:** 10.3390/cells9051264

**Published:** 2020-05-20

**Authors:** Sebastian Klobuch, Kathrin Hammon, Sarah Vatter-Leising, Elisabeth Neidlinger, Michael Zwerger, Annika Wandel, Laura Maria Neuber, Bernhard Heilmeier, Regina Fichtner, Carina Mirbeth, Wolfgang Herr, Simone Thomas

**Affiliations:** 1Department of Internal Medicine III, University Hospital Regensburg, 93042 Regensburg, Germany; kathrin.hammon@ukr.de (K.H.); Sarah.Vatter@web.de (S.V.-L.); Elisabeth.Neidlinger@stud.uni-regensburg.de (E.N.); michael_zwerg_er@gmx.de (M.Z.); Annika.Wandel@stud.uni-regensburg.de (A.W.); Laura-Maria.Neuber@stud.uni-regensburg.de (L.M.N.); regina.fichtner@ukr.de (R.F.); carina.mirbeth@ukr.de (C.M.); wolfgang.herr@ukr.de (W.H.); 2Regensburg Center for Interventional Immunology, University of Regensburg, 93042 Regensburg, Germany; 3Department of Oncology and Hematology, Hospital Barmherzige Brueder, 93049 Regensburg, Germany; Bernhard.Heilmeier@barmherzige-regensburg.de

**Keywords:** HLA-DP, allogeneic stem cell transplantation, TCR gene therapy, graft versus leukemia reaction, adoptive immunotherapy

## Abstract

HLA-DPB1 antigens are mismatched in about 80% of allogeneic hematopoietic stem cell transplantations from HLA 10/10 matched unrelated donors and were shown to be associated with a decreased risk of leukemia relapse. We recently developed a reliable in vitro method to generate HLA-DPB1 mismatch-reactive CD4 T-cell clones from allogeneic donors. Here, we isolated HLA-DPB1 specific T cell receptors (TCR DP) and used them either as wild-type or genetically optimized receptors to analyze in detail the reactivity of transduced CD4 and CD8 T cells toward primary AML blasts. While both CD4 and CD8 T cells showed strong AML reactivity in vitro, only CD4 T cells were able to effectively eliminate leukemia blasts in AML engrafted NOD/SCID/IL2Rγc^−/−^ (NSG) mice. Further analysis showed that optimized TCR DP and under some conditions wild-type TCR DP also mediated reactivity to non-hematopoietic cells like fibroblasts or tumor cell lines after HLA-DP upregulation. In conclusion, T cells engineered with selected allo-HLA-DPB1 specific TCRs might be powerful off-the-shelf reagents in allogeneic T-cell therapy of leukemia. However, because of frequent (common) cross-reactivity to non-hematopoietic cells with optimized TCR DP T cells, safety mechanisms are mandatory.

## 1. Introduction

In allogeneic hematopoietic stem cell transplantation (alloHSCT), donor T cells can mediate the beneficial graft versus leukemia (GvL) effect. Donor T cells recognize minor histocompatibility antigens on patient cells and the magnitude and diversity of this response determines the selectivity of the GvL effect [1]. However, donor T cells can also cause one major reason of morbidity and mortality after alloHSCT, which is graft versus host disease (GvHD). In addition, in some patients the GvL effect is not sufficient to eradicate patients’ leukemia. Therefore, more effective strategies are needed to treat patients with recurrent disease after alloHSCT [2].

One possible strategy is the transfer of genetically engineered T cells, which specifically recognize patients’ leukemia cells [3]. However, targeting an antigen which is expressed exclusively on malignant cells and not on healthy tissues is still one of the major hurdles for cell therapeutic applications in the field. In case of B cell malignancies such as acute lymphoblastic leukemia or diffuse large B-cell lymphoma, targeting CD19 by chimeric antigen receptor (CAR) T cells showed impressive clinical responses [4]. In contrast, suitable target antigens with broad application across patients with myeloid malignancies like acute myeloid leukemia (AML) are lacking so far.

In this study we aimed at targeting allo-HLA-DPB1 alleles with T cells from allogeneic donors. Since class II HLA-DPB1 antigens are mismatched in about 80% of HLA-A, -B, -C, -DR, -DQ (i.e., 10 of 10) matched unrelated donor transplantations [5], it is feasible to target the mismatched HLA-DPB1 allele on patient cells with donor T cells. In previous studies, certain HLA-DPB1 mismatch constellations were shown to be beneficial in terms of lower risk of leukemia relapse after alloHSCT [6,7,8,9,10]. More recently, HLA-DPB1 was found to be downregulated on leukemia blasts in a significant fraction of patients who have relapsed after transplantation [11,12], implicating an important role of HLA-DPB1 as a target antigen for GvL reactivity but also its potential involvement in immune escape after alloHSCT.

We previously reported on an efficient in vitro method to isolate and propagate HLA-DPB1 reactive T cell clones from healthy blood donors [13,14]. Those T cell clones showed highly specific recognition and lysis of primary AML blasts in vitro as well as in a xenograft mouse model [13]. Herein, we isolated the T cell receptor (TCR) genes of different T cell clones recognizing allo-HLA-DPB1*04:01 or allo-HLA-DPB1*03:01. TCR coding sequences were then genetically optimized and transferred into donor T cells, and detailed analysis of the recognition pattern against hematopoietic and non-hematopoietic target cells was performed. Of note, we also analyzed adoptively transferred TCR modified T cells in the AML xenograft mouse model.

## 2. Materials and Methods

### 2.1. Cell Lines and Primary Material

Primary AML blasts were isolated from peripheral blood of patients before initiation of leukemia therapy. Peripheral blood mononuclear cells (PBMC) were isolated from leukocyte reduction system cones of healthy donors by standard Ficoll density separation. All subjects gave their informed consent for blood donation before they participated in the study. The study was conducted in accordance with the Declaration of Helsinki, and the protocols were approved by the Ethics Committee of the University Hospital Regensburg (permission numbers 05-097 and 13-101-0240). Epstein–Barr virus (EBV)-transformed B-lymphoblastoid cell lines (LCL) and primary fibroblasts were generated and cultured as previously described [15]. Tumor cell lines were purchased from ATCC and cultured according to recommended procedures. K562 cell line (ATCC, Manassas, VA, USA) lack HLA-class I and -II surface expression and was originally established from the pleural effusion of a patient with chronic myelogenous leukemia in blast crisis.

### 2.2. TCR Sequences and Cloning

TCR sequences from CD4 T cell clones were identified as reported [14]. TCR coding genes were then ordered and codon optimized by GeneArt (Thermo Fisher Scientific, Waltham, MA, USA). Murinization of the TCR constant regions was performed as described by others [16]. Coding sequences of TCRα and TCRβ genes were either cloned into multiple cloning site of the drug-selectable retroviral vector pMX (BioCat, Heidelberg, Germany) linked by a self-cleaving F2A element [17] for retroviral transfer or were inserted separately into the pGEM4Z vector for RNA transfer as previously described [18]. In-vitro-transcription of TCR encoding RNA (IVT-RNA) was performed with T7 RNA-polymerase using the mMESSAGE mMACHINE™ T7 ULTRA Transcription Kit (Thermo Fisher Scientific, Waltham, MA, USA) according to manufacturing instructions.

### 2.3. Flow Cytometry and Antibodies

Flow cytometry was performed on BD FACS Calibur (BD Biosciences, Heidelberg, Germany) and analyzed by FlowJo 10.4.2 software (BD Biosciences, Heidelberg, Germany). The following fluorochrome-labeled monoclonal antibodies (mAb) were used: anti-human CD3 (UCHT1), CD4 (RPA-T4), CD8 (RPA-T8), CD33 (WM53), CD45 (HI30), anti-mouse TCRβ chain (H57-597) (all BD Biosciences, Heidelberg, Germany), and anti-human TCR vβ 13.2 (H132; Beckman Coulter, Brea, CA, USA). Anti-human HLA-DP (B7/21; Leinco Technologies Inc., St. Louis, MO, USA) was labeled with a secondary goat-anti-mouse FITC mAb (Thermo Fisher Scientific, Waltham, MA, USA).

### 2.4. Retroviral Transduction and RNA Electroporation of T Cells

CD4 and CD8 T cells were positively isolated from PBMC of healthy donors by standard MACS cell separation (Miltenyi Biotec, Bergisch Gladbach, Germany).

Retroviral transfer of TCR genes was performed using virus-containing cell-culture supernatant from amphotropic retroviral packaging cell line Platinum-A (BioCat, Heidelberg, Germany) transfected with TransIT-LT1 Transfection Reagent (Mirus Bio, Madison, WI, USA) and 11.5 µg pMX vector DNA. Virus containing supernatant was collected after 48 and 72 h. Before transduction, T cells were activated for three days by anti-CD3/CD28 beads (Thermo Fisher Scientific, Waltham, MA, USA) and 100 IU/mL recombinant human (rh) IL-2 (Prometheus Laboratories Inc., San Diego, CA, USA) in AIM-V medium (Thermo Fisher Scientific, Waltham, MA, USA) supplemented with 10% human serum. Subsequently, TCR expressing T cells were drug-selected with puromycin for 3 days and expanded using anti-CD3/CD28 beads in AIM-V medium supplemented with 50 IU/mL rh IL-2, 5 ng/mL rh IL-7 (Miltenyi Biotec, Bergisch Gladbach, Germany), 5 ng/mL rh IL-15 (Bio-Techne, Minneapolis, MI, USA), and 10% human serum (AIM-V^cytokine^).

RNA transfer of the TCR was performed by electroporation as described [18]. Briefly, T cells were stimulated and expanded over 10–14 days in vitro by anti-CD3/CD28 beads (Thermo Fisher Scientific, Waltham, MA, USA) in AIM-V^cytokine^ medium. Electroporation of pre-stimulated T cells was performed in a 4-mm cuvette with the GenePulser Xcell system (Bio-Rad, Munich, Germany) applying a square wave pulse of 500 V, 5 ms, to 5–10 × 10^6^ cells in the presence of 10 µg IVT-RNA for each TCR chain.

### 2.5. In Vitro Assays of TCR Modified T Cells

RNA transfected T cells were tested for their activity after overnight incubation. Standard 20 h Interferon (IFN)-γ ELISpot and 4 to 5 h ^51^Chromium (Cr)-release assays were performed in duplicate wells as described [19]. IFN-γ (500 IU/mL, Thermo Fisher Scientific, Waltham, MA, USA) pretreatment was performed over 4 days (primary fibroblasts and tumor cell lines) or 24 h (primary AML blasts) before testing. RNA electroporation of tumor cell lines with HLA-DPA1 and HLA-DPB1 encoding IVT-RNA was performed 16–20 h before testing and cells were electroporated with 10 µg RNA for each HLA-DP chain applying a square wave pulse of 400 V, 5 ms as previously reported [13].

### 2.6. AML Xenograft Mouse Model

Animal experiments were approved by the local regulatory authorities according to German Federal Law (permission number 54-2532.1-42/13). NOD.Cg-Prkdc^scid^IL2rg^tmWjl^/Sz (NSG) mice (Jackson Laboratory, Bar Harbor, Maine, USA) were bred under SPF conditions. Engraftment of primary human AML blasts and adoptive transfer of TCR-modified T cells were performed as reported [13]. Briefly, 6- to 12-week-old female NSG mice were sub-lethally γ-irradiated with 1.5 Gy on the day before intravenous injection of 4 × 10^6^ primary AML blasts. On day 21 after engraftment, 1 × 10^7^ retrovirally transduced T cells were intravenously transferred together with 1000 IU/mL rh IL-2 and 20 µg Fc-IL-7 (Merck, Darmstadt, Germany). On day 7 after T-cell injection, AML infiltration was investigated in bone marrow of the animals by flow cytometry.

### 2.7. Statistical Analysis

Statistical analysis was performed with GraphPad Prism 8.1.1 (GraphPad Software, San Diego, CA, USA), and *p*-values were calculated by Kruskal–Wallis test with Dunn’s correction for multiple comparisons. Differences were considered statistically significant for *p* values of < 0.05.

## 3. Results

### 3.1. TCR DP04 Triggers Specific Recognition and Lysis of AML Blasts by CD4 and CD8 T Cells

We previously described a CD4 T cell clone (clone 11C12) recognizing allogeneic HLA-DPB1*04:01 expressing cells [13]. Because this T cell clone induced effector function in a CD4-independent manner (i.e., after blocking the engagement of the CD4 co-receptor and the HLA-DP molecule on the target cell by a CD4 blocking mAb) we assumed that the TCR could be used for the redirection of both CD4 and CD8 T cells [14]. We therefore isolated the TCRα (TRAV13-2, nomenclature according to ImMunoGeneTics (IMGT) [20]) and TCRβ (TRBV6-2, IMGT) sequences from this T cell clone and murinized the TCR by exchanging the human constant domains by their murine counterparts to increase expression and to promote preferential pairing of transferred TCRα and β chains [16]. This TCR is further referred to as TCR DP04_chim_. Figure 1A shows a representative example of TCR DP04_chim_ expression in pre-stimulated T cells of an HLA-DPB1*04:01 negative healthy donor 16–20 h after electroporation of TCRα and β encoding RNA, which leads to high surface expression of the TCR (> 96% vβ 13.2 positive cells) in CD4 as well as CD8 T cells. In contrast, T cells transfected without RNA (Mock) stained only slightly positive, representing naturally expressed TCRs of the same TCR vβ 13.2 subfamily.

Next, we analyzed recognition of primary AML blasts by IFN-γ ELISpot assay (Figure 1B). TCR DP04_chim_ modified CD4 T cells showed highly specific IFN-γ secretion against HLA-DPB1*04:01 positive AML blasts (AML111, AML121, AML128) from individual patients and EBV-LCL (Figure 1B, left panel). This recognition was TCR-specific as indicated by its absence after Mock (w/o RNA) transfection of the T cells as well as after co-incubation with HLA-DPB1*04:01 negative target cells (AML110 and EBV-LCL) (Figure 1B). In CD8 T cells, TCR DP04_chim_ transfected cells showed strong IFN-γ production upon co-incubation with HLA-DPB1*04:01 positive AML sample 111 and EBV-LCL, but only weak (AML121) or no (AML128) recognition of other HLA-DPB1*04:01 positive AML blasts (Figure 1B, right panel). This recognition could only be slightly enhanced by IFN-γ pretreatment, which is known to enhance HLA class II expression on AML blasts (Figure 1B and Appendix A) [13]. We then analyzed cytolytic activity of TCR DP04_chim_ expressing CD4 and CD8 T cells against primary AML blasts. Here, both TCR DP04_chim_ redirected CD4 and CD8 T cells showed highly specific lysis of HLA-DPB1*04:01 expressing AML blasts (AML111, AML121, AML128), while sparing an HLA-DPB1*04:01 negative sample (AML110) (Figure 1C). Notably, lysis activity of CD8 T cells exceeded that of CD4 cells. Cytolytic CD4 T cells have already been described in the context of GvL reponses [21,22] and we have previously identified the exocytosis pathway of cytotoxic granules (granzyme A and B, perforin) as predominant lysis mechanism of HLA-DP specific CD4 T cells [13].

### 3.2. TCR DP04_chim_ Modified CD4 T Cells Effectively Eliminate Human AML Blasts in NSG Mice

After characterization of TCR DP04_chim_ modified T cells in vitro, we tested their ability to control primary human AML blasts in immunodeficient NSG mice at a time point when bone marrow infiltration reached 1–5%. In detail, sublethally irradiated NSG mice were injected with 4 x 10^6^ cells of AML167 (HLA-DPB1*04:01/17:01). On day 21, 1 × 10^7^ TCR DP04_chim_ retrovirally transduced CD4 T cells were intravenously transferred into the mice and leukemia burden as well as T cell frequencies in bone marrow were analyzed seven days later (i.e., day 28). In contrast to untreated mice (mean 3.0%, SD 1.6) or animals injected with CD4 T cells expressing a CMVpp65 specific control TCR (mean 5.4%, SD 3.3), CD4 TCR DP04_chim_ modified T cells almost completely eliminated AML167 blasts (mean 0.2%, SD 0.2) in bone marrow of the mice as assessed by flow cytometry (Figure 2A). Transferred T cells were found in both TCR DP04_chim_ and TCR control treated groups (Figure 2B). We also performed in vivo experiments with TCR DP04_chim_ transduced CD8 T cells. However, AML clearance could not be reliably reproduced throughout multiple experiments (Appendix A), which could be due to the absence of other human cytokines like IL-15 or IL-21 in our model, which have been show to promote CD8 T cell engraftment in NSG mice [23]. This is supported by a lower engraftment of CD8 versus CD4 TCR modified T cells in our xenograft model (Appendix A). Signs of xenogeneic GvH reactivity like weight loss, change of fur texture, or kyphosis were not detected in both CD4 and CD8 treated mice.

### 3.3. Reactivity of TCR DP04_chim_ is Restricted to HLA-DPB1*04:01 Positive Cells, but not to Hematopoietic Cells

We further investigated the HLA-DPB1 specificity of TCR DP04_chim_ and chose CD4 T cells for cross-reactivity experiments to maintain the contribution of the CD4 co-signal. For this we transfected K562 cells that do not express HLA class-I and -II molecules on the cell surface and are described as optimal antigen presenting cell with low allo-recognition [24] with a broad panel of HLA-DPB1 alleles and their most frequently genetically linked HLA-DPA1 allele [25]. HLA-DP expression was confirmed by flow cytometry (Appendix A). Upon co-incubation only K562 cells expressing the HLA-DPB1*04:01 allele and the very similar HLA-DPB1*04:02 allele (both co-expressed with the HLA-DPA1*01:03 allele) were able to trigger significant IFN-γ spot production by TCR DP04_chim_-modified CD4 T cells (Figure 3A).

Constitutive expression of HLA class II molecules is restricted to cells of hematopoietic origin under non-inflammatory conditions. However, in the presence of a pro-inflammatory cytokine milieu (e.g., during infection), HLA class II expression is also upregulated on other cell types, which might trigger an HLA-DP specific GvH reactivity [26]. Therefore, we investigated the ability of TCR DP04_chim_ expressing T cells to recognize primary human fibroblasts from individual donors that do not express HLA-class II antigens during non-inflammatory conditions and can be used as surrogate cells for GvH reactivity [13,27]. While fibroblasts under non-inflammatory conditions were almost not recognized (< 12 IFN-γ spots per 15.000 T cells) by TCR DP04_chim_-expressing CD4 and CD8 T cells (Figure 3B), pretreating them with IFN-γ elicited an HLA-DPB1*04:01 specific IFN-γ secretion in both TCR DP04_chim_ modified CD4 and CD8 T cells (> 500 IFN-γ spots per 15.000 T cells, Figure 3B). In contrast, Mock transfected CD4 or CD8 T cells did not react with either untreated or IFN-γ-pretreated fibroblasts, respectively (Figure 3B).

### 3.4. HLA-DPB1*03:01 Specific TCR Recognizes AML Blasts, but not Fibroblasts under Physiological Conditions

Because TCR DP04_chim_ showed strong reactivity to IFN-γ pretreated non-hematopoietic cells (i.e., fibroblasts), we screened for other HLA-DP reactive CD4 T cell clones potentially restricted to hematopoietic cells generated in our T cell stimulation protocol [13,14]. From cultures co-incubating CD45RA-selected CD4 T cells and autologous dendritic cells expressing allogeneic HLA-DPA1*01:03/DPB1*03:01 upon RNA transfection, we were able to isolate an HLA-DPB1*03:01 specific CD4 T cell clone (i.e., clone 11G1) with reactivity to primary AML blasts [14], but most notably, there was no reactivity to primary fibroblasts even after HLA-DP induction upon IFN-γ pretreatment (Appendix A). Again, TCRα (TRAV8–3, IMGT) and TCRβ (TRBV11-3, IMGT) sequences were isolated and cloned either in their wild-type form or with murinized constant TCR domains into the pGEM4Z vector (further referred to as TCR DP03_WT_ and TCR DP03_chim_, respectively).

To confirm reactivity of the CD4 T cell clone 11G1 we next analyzed the recognition pattern of CD4 and CD8 T cells expressing TCR DP03_WT_ upon RNA transfection. In contrast to the HLA-DPB1*04:01 specific TCR, TCR DP03_WT_ modified CD4 and CD8 T cells showed almost no IFN-γ secretion upon co-incubation with either untreated (< 11 IFN-γ spots per 15.000 T cells) or IFN-γ pretreated HLA-DPB1*03:01 positive fibroblasts (< 45 IFN-γ spots per 15.000 T cells) in ELISpot assay, whereas HLA-DPB1*03:01 positive EBV-LCL were recognized (Figure 4A, grey bars). However, the detected background reactivity against HLA-DPB1*03:01 negative EBV-LCL and of Mock T cells might in part be explained by EBV-specific reactivity triggered by endogenous TCRs, particularly with CD4 T cells. In contrast, the murinized TCR DP03_chim_ showed recognition of IFN-γ pretreated HLA-DPB1*03:01 positive fibroblasts when expressed in CD4 T cells (up to 278 IFN-γ spots per 15.000 T cells, Figure 4A, left panel, black bars). In TCR DP03_chim_-transfected CD8 T cells, much lower IFN-γ spot formation was observed (up to 86 IFN-γ spots per 15.000 T cells, Figure 4A, right panel, black bars). The recognition of HLA-DPB1*03:01 antigen on pretreated fibroblasts by TCR DP03_chim_ might be due to a higher expression of murinized TCR in comparison to full-length human TCR chains [16]. However, TCR expression analysis of TCR DP03_WT_ and TCR DP03_chim_ was not possible because a TCR vβ 21.2 specific monoclonal antibody is not commercially available. CD4 and CD8 T cells transfected with the murinized TCR DP03_chim_ showed a robust TCR expression (> 95% in CD4 and CD8 T cells) as detected by cytofluorometric analysis with an antibody that specifically binds to the murine constant TCRβ domain (Appendix A).

We then analyzed the ability of TCR DP03–modified T cells to specifically recognize primary AML blasts in IFN-γ ELISpot (Figure 4B) and ^51^Cr-release-assay (Figure 4C). Both, TCR DP03_WT_ and TCR DP03_chim_ expressing T cells specifically secreted IFN-γ and lysed AML blasts upon co-incubation with samples from individual HLA-DPB1*03:01 positive patients (Figure 4B,C). Again, and similar to TCR DP04_chim_ expressing T cells, lysis efficacy of TCR DP03_WT_ and TCR DP03_chim_ modified CD8 T cells exceeded that of CD4 T cells (Figure 4C). In addition, murinized TCR DP03_chim_ induced higher IFN-γ spot production compared to TCR DP03_WT_, especially in CD8 T cells (Figure 4B, right panel).

Overall, the presumed enhancement of TCR DP03_chim_ expression level in comparison to that of TCR DP03_WT_ increased reactivity to primary AML blasts and increased the HLA-DPB1*03:01 specific cytokine response against IFN-γ pretreated fibroblasts, particularly in CD4 T cells.

### 3.5. Expression Level of HLA-DPB1 Determines Recognition of Target Cells by TCR DP03

To analyze the recognition of other non-hematopoietic cell types by TCR DP03 modified T cells, we transfected different tumor cell lines from human (SK29mel, Saos, SW480) and primate origin (Cos-7) with HLA-DPA1*01:03 and -DPB1*03:01 alleles resulting in high HLA-DP surface expression (Appendix A). As expected from experiments shown in Figure 4A, TCR DP03_chim_ transfected CD4 and CD8 T cells showed strong recognition (>400 spots per 15.000 CD4 and CD8 T cells, respectively) of all tested HLA-DPB1*03:01 transfected tumor cell lines (Figure 5A). Albeit to a lesser extent, TCR DP03_WT_ modified CD4 and CD8 T cells also showed reactivity to HLA-DPB1*03:01 transfected cell lines, whereas untransfected counterparts were not recognized by TCR DP03_WT_ CD4 and CD8 T cells (Figure 5A).

To further analyze the different recognition patterns induced by TCR DP03_WT_, we compared the recognition of human primary fibroblasts expressing HLA-DP antigen upon IFN-γ pretreatment or HLA-DPB1 transfection. As shown in Figure 5B, electroporation of fibroblasts with HLA-DP coding RNA induced IFN-γ secretion in both TCR DP03_WT_ and TCR DP03_chim_ expressing CD4 T cells (253 and 541 IFN-γ spots, respectively) and to a lower extent in CD8 T cells (26 and 374 IFN-γ spots, respectively). In contrast IFN-γ pretreated fibroblasts only induced IFN-γ spot formation in CD4 TCR DP03_chim_ but not TCR DP03_wt_ expressing T cells (21 and 355 IFN-γ spots, respectively), as observed in experiments shown in Figure 4A. CD8 T cells either transfected with WT or chimeric TCR did not react with IFN-γ pretreated fibroblasts at all. 

To compare surface expression of HLA-DP antigen, we analyzed IFN-γ pretreated and HLA-DP transfected fibroblasts by flow cytometry. Here, we observed that electroporation of fibroblasts with HLA-DPA1*01:03/DPB1*03:01 coding RNA led to an HLA-DP surface expression that is about one log higher (median fluorescence intensity (MFI_x_) 160) compared to IFN-γ pretreatment (MFI_x_ 22) (Figure 5C, left panel). We observed similar results in two human tumor cell lines (melanoma SK5mel, colon carcinoma HCT116), even though the upregulation of HLA-DP by IFN-γ resulted only in a small shift of HLA-DP surface expression (Figure 5C, middle and right panel). Most notably, whereas IFN-γ pretreatment did not lead to HLA-DP expression levels which are sufficiently high for recognition of the cell lines by TCR DP03_WT_, HLA-DP electroporation again resulted in clear activation and IFN-γ secretion by TCR DP03_WT_ modified CD4 T cells (Figure 5E). 

However, CD8 T cells only recognized HLA-DPB1*03:01-transfected tumor cells upon expression of TCR DP03_chim_ but not TCR DP03_WT_ (Figure 5E)_._ In line with the data shown in Figure 4, HLA-DPB1*03:01 positive primary AML blasts were recognized by both TCR DP03_WT_ and to a greater extent by TCR DP03_chim_ CD4 and CD8 T cells, respectively (Figure 5E). Primary AML blasts showed an intermediate to high expression level of HLA-DP under non-treated conditions when compared to non-hematopoietic cells upon IFN-γ pretreatment or HLA-DP electroporation, respectively (Figure 5D).

In conclusion, the surface expression level of HLA-DPB1 in target cells, among other factors like TCR expression level, is a very important determinant that strongly influences recognition by TCR DP03 redirected T cells.

## 4. Discussion

Adoptive T cell therapy has recently shown impressive clinical efficacy, especially with CAR modified T cells [4]. However, targeting myeloid derived malignancies like AML remains challenging due to the lack of appropriate target antigens. In the setting of alloHSCT, the spectrum of possible targets is broadened by naturally occurring histocompatibility antigens that are mismatched between patient and donor. Therefore, adoptive immunotherapy with allo-HLA-DPB1 reactive donor T cells aims to use these genetic differences for therapeutic purpose in order to strengthen the GvL effect in patients after alloHSCT [13]. This is even underlined by an increasing body of literature providing evidence for HLA-DPB1 as an important target of the GvL immune response in patients [28,29,30,31].

In the present study, we show that targeting of allo-HLA-DPB1 alleles by TCR modified T cells is feasible and could result in strong HLA-DPB1-specific reactivity of CD4 and CD8 T cells against human primary AML samples. However, because of the induction of HLA class II expression on non-hematopoietic cells by pro-inflammatory cytokines, TCR DP reactive TCRs might also induce an undesirable HLA-DP specific GvH reactivity under inflammatory conditions (e.g., viral infection [26]) therefore hampering clinical application. In our study, TCR DP04_chim_ broadly recognized HLA-DPB1*04:01 positive cells irrespectively of their origin (Figure 3B and data not shown). As this TCR was generated in an HLA-DP mismatch setting, it was previously reported that allogeneic T cell clones show higher avidity, but also a higher rate of off-target recognition [32]. We speculate that this TCR recognizes a peptide that is ubiquitously expressed and restricted to HLA-DPA1*01:03/DPB1*04:01, however, experimental data on this are lacking. Identification of the natural peptide sequence would require to purify HLA-DPA1*01:03/DPB1*04:01 associated peptide ligands followed by complex liquid chromatography steps and sequencing of the T-cell recognized peptide by mass spectrometry analysis [33,34]. Alternatively, the recognition pattern of TCR DP04 may be determined by the recognition of a peptide terminus instead of a full-length peptide as the binding groove of HLA-class II molecules is open on both sides and particularly the N-terminal extension can play a major role in the TCR interaction [35].

For clinical application of HLA-DP specific TCRs it would be ideal to find TCRs that recognize HLA-DP restricted peptides specifically expressed by hematopoietic cells. Nevertheless, TCR DP03_WT_ induced a robust recognition (i.e., IFN-γ secretion and cytolytic activity) of primary AML blasts (Figure 4B,C), but most importantly, lacked reactivity against fibroblasts when HLA-DP was physiologically induced by IFN-γ pretreatment (Figure 4A). However, when HLA-DP was expressed at extraordinary high levels by transfection, TCR DP03_WT_ also reacted with cells of non-hematopoietic origin (Figure 5A,B). This finding suggests that TCR DP03_WT_ might also recognize a peptide ubiquitously expressed by both hematopoietic and non-hematopoietic cells, however with stronger expression of the peptide or/and the restricting HLA in former tissue. Based merely on our reactivity data, the TCR DP03_WT_ appears to be a more suitable receptor for therapeutic purpose compared to TCR DP03_chim_ or TCR DP04_chim_. 

It was already shown by others that HLA-DP specific T cells with reactivity pattern restricted to hematopoietic cells can indeed be generated [36]. Therefore, extensive screening for such TCRs seems to be a reasonable strategy. Another possibility for clinical application of HLA-DP specific TCRs is the use of safety switches like inducible Caspase 9, truncated epidermal growth factor receptor (EGFR) or alternative on-switch modules that allow for a specific depletion of TCR modified T cells or a specific control of the T cell activity [37,38,39,40].

Our data also highlight that increasing the affinity of the TCR or moreover the avidity between the T cell and the target cell, which is dictated by a combination of the affinity of its TCR for a defined HLA/peptide complex and the number of TCR surface molecules, should be carried out carefully, because it also increases the risk of unwanted reactivity as shown for TCR DP03_chim_ (Figure 4A and Figure 5A) and other tumor-specific TCRs [41]. Indeed, augmenting the density of TCR surface expression by murinization was shown to overcome a natural low avidity between TCR transgenic cells and tumor cells [16,42]. Another example for cross-reactivity induced by affinity-enhancement was shown for a MAGE-A3-specific TCR [43]. Here, amino acid substitutions within the TCRα complementary determining region (CDR) 2 led to fatal cross-reactivity against an unrelated peptide derived from the cardiomyocyte expressed protein titin.

In addition, in vitro tests have limitations concerning their ability to determine and predict safety in patients. This is for example shown by the differential HLA-DP expression level after RNA electroporation and IFN-γ pretreatment of target cells, which leads to different T cell recognition patterns (Figure 5B). It is also unknown, if the HLA-DP expression levels occurring in patients under inflammatory conditions are sufficient to activate TCR DP03_WT_ modified T cells and to induce damage of healthy cells under these circumstances. The complex regulation and different HLA-DP expression levels on various cell types is also underlined by an interesting study published by Petersdorf et al. Theses authors showed that a higher expression level of HLA-DP determined by a genetic variant in the regulatory region of HLA-DPB1 increases the risk for GvHD after alloHSCT [6].

In summary, we show in this study that the genetic transfer of HLA-DPB1 specific TCRs into CD4 and CD8 T cells mediates strong immune reactivity including IFN-γ production and cytolytic activity to primary human AML blasts. By screening various TCRs, we found receptors with reactivity clearly exceeding to hematopoietic versus non-hematopoietic cells. Even with these prioritized TCRs, the genetic modification of TCR to enhance anti-leukemic activity as well as the variable expression level of HLA-DPB1 in non-hematopoietic tissues still determines (unwanted) reactivity to non-hematopoietic cells. Taking this into consideration, HLA-DPB1 reactive TCRs appear to be promising tools to strengthen the GvL effect after alloHSCT. However, for clinical application further investigations have to be performed to improve the hematopoiesis restricted specificity and safety of this approach.

## Figures and Tables

**Figure 1 cells-09-01264-f001:**
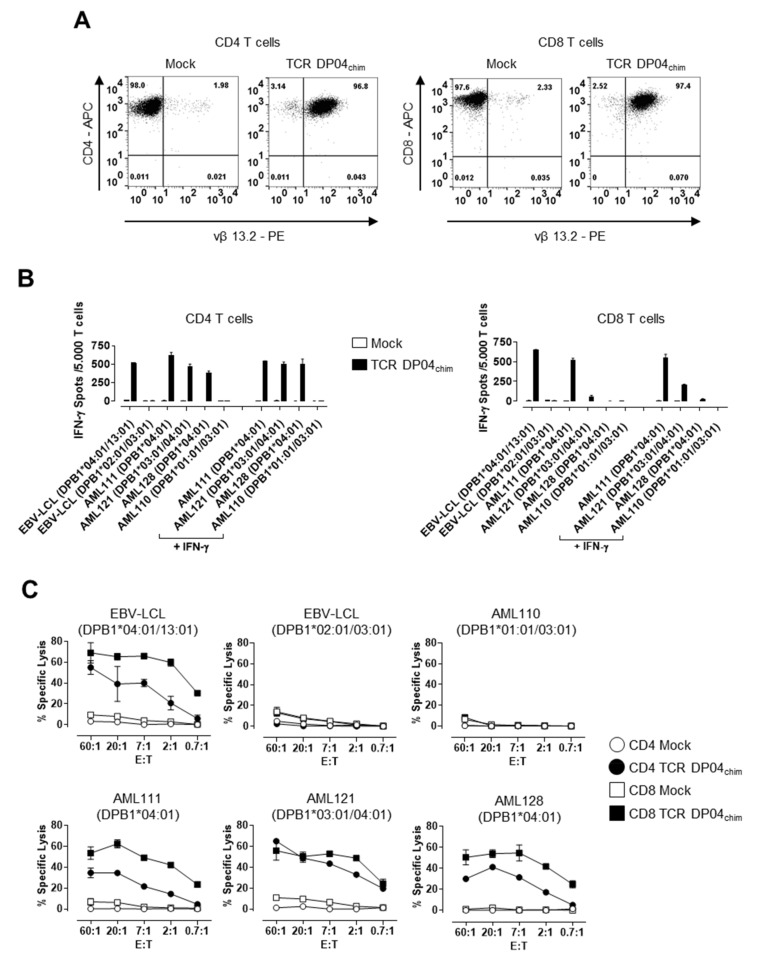
T cell receptor (TCR) expression and reactivity of TCR DP04_chim_ redirected T cells. (**A**) Immunomagnetically selected and prestimulated human CD4 (left panels) and CD8 T cells (right panels) from an HLA-DPB1*04:01 negative healthy donor were transfected with TCR DP04_chim_ coding RNA (CD4 TCR DP04_chim_ and CD8 TCR DP04_chim_) or without RNA (CD4 Mock and CD8 Mock) and analyzed after 16–20 h by flow cytometry for expression of CD4, CD8, as well as of TCR DP04_chim_ using TCR vβ 13.2 subfamily specific mAb. (**B**) IFN-γ spot formation and (**C**) cytolytic activity of TCR DP04_chim_- and Mock-transfected CD4 and CD8 T cells upon incubation with HLA-DPB1*04:01 positive acute myeloid leukemia (AML) blasts from individual patients and EBV-LCL or, as controls with HLA-DPB1*04:01 negative target cells at an effector-to-target cell ratio (E:T) of (**B**) 0.1:1 or (**C**) as indicated. AML blasts in (**B**) were either left untreated or pretreated with 500 IU/mL IFN-γ for 24 h before testing. Standard deviation of mean is shown of two technical replicates. All experiments in Figure 1 are representative of one T cell donor out of three.

**Figure 2 cells-09-01264-f002:**
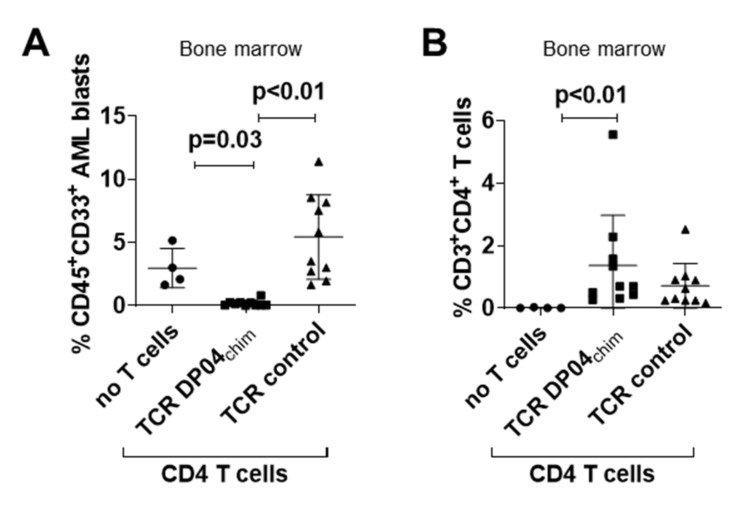
Anti-leukemic activity of TCR DP04_chim_ modified CD4 T cells in AML-engrafted NSG mice. Sublethally irradiated (1.5 Gy) NSG mice were intravenously injected with 4 × 10^6^ primary blasts of AML167 (HLA-DPB1*04:01/17:01). On day 21, when leukemia engraftment reached a level of 1%–5% in bone marrow as confirmed in identically treated control mice, 1 × 10^7^ CD4 T cells retrovirally transduced with TCR DP04_chim_ (*n* = 10) or a control TCR (CMVpp65/HLA-A*02:01-specific; *n* = 10) were intravenously injected along with single doses of rh IL-2 (1000 IU) and FcIL-7 (20 µg). Animals in the control group without T cells (*n* = 4) only received rh IL-2 and Fc-IL-7. On day 28 (i.e., seven days after T cell transfer), (**A**) AML burden (CD45^+^ CD33^+^ cells) and (**B**) T cell frequencies (CD3^+^ CD4^+^ cells) were analyzed in bone marrow. Results are pooled from two independent experiments. Symbols represent individual mice and horizontal bars mark mean values with standard deviation. *p*-values were calculated by Kruskal–Wallis test with Dunn’s correction for multiple comparisons, *p* < 0.05 considered significant.

**Figure 3 cells-09-01264-f003:**
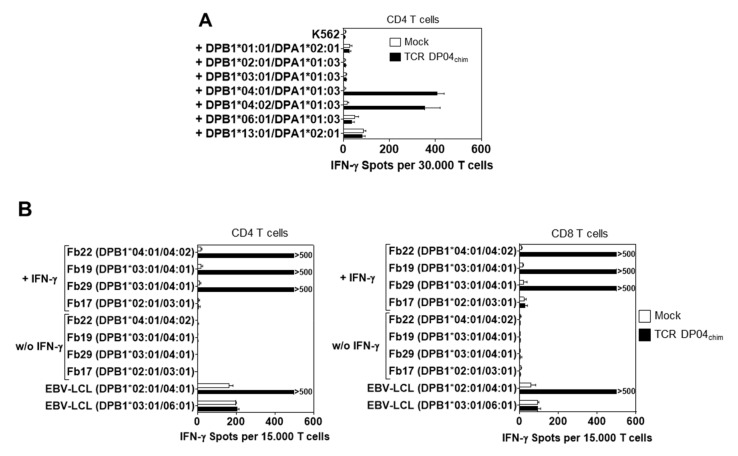
Reactivity of TCR DP04_chim_ redirected T cells against different HLA-DP alleles. (**A**) K562 were transfected with RNA encoding for different HLA-DPB1 chains and their most common associated HLA-DPA1 chains. Pre-stimulated CD4 T cells were transfected with TCR DP04_chim_ coding RNA (CD4 TCR DP04_chim_) or without RNA (CD4 Mock) and IFN-γ spot formation was tested upon incubation with HLA-DP transfected K562 cells at an E: T ratio of 0.6:1. The figure depicts spot numbers from three different HLA-DPB1*04:01 negative healthy T cell donors and standard deviation of mean is shown. (**B**) Pre-stimulated human CD4 (left panel) and CD8 T cells (right panel) from an HLA-DPB1*04:01 negative healthy donor were transfected with TCR DP04_chim_ coding RNA or without RNA (Mock) and incubated with untreated or IFN-γ pretreated primary fibroblasts from HLA-DPB1*04:01 positive and negative donors at an E: T ratio of 0.3:1. IFN-γ spot production was measured after 16–20 h. EBV-LCL from HLA-DPB1*04:01 positive and negative donors served as controls. Standard deviation of mean of two technical replicates is shown and experiment was performed with three different T cell donors, from which one representative is depicted.

**Figure 4 cells-09-01264-f004:**
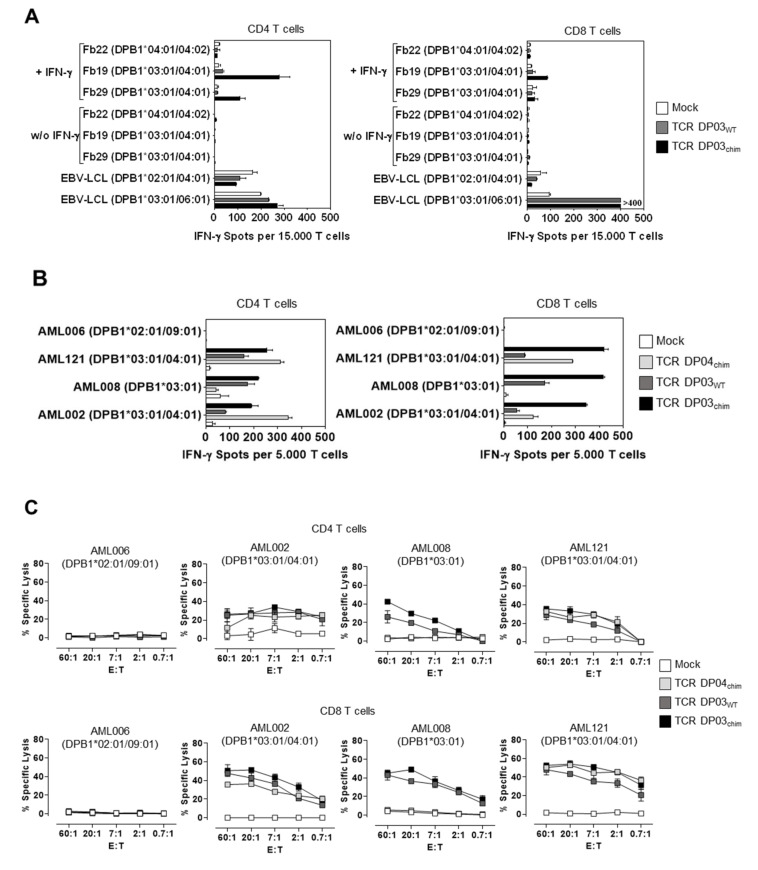
Reactivity of TCR DP03 against primary fibroblasts and AML blasts. (**A**) Pre-stimulated human CD4 (left panel) and CD8 T cells (right panel) from an HLA-DPB1*03:01 negative healthy donor were transfected with TCR DP03_WT_ or TCR DP03_chim_ coding RNA or without RNA (Mock). T cells were incubated with untreated or IFN-γ pretreated fibroblasts from HLA-DPB1*03:01 positive and negative donors in an E: T ratio of 0.3:1 to analyze IFN-γ spot production in ELISpot assay. EBV-LCL from HLA-DPB1*03:01 positive and negative donors served as positive and negative controls. Standard deviation of mean of IFN-γ spot formation of two technical replicates is shown. (**B**) IFN-γ spot formation and (**C**) cytolytic activity of indicated CD4 and CD8 T cell populations. Primary AML blasts from HLA-DPB1*03:01 positive and negative donors were used as target cells in an E: T ratio of 0.1:1 (**B**) or as indicated (**C**). Standard deviation of mean is shown for two technical replicates. Data are shown from one experiment with a representative T cell donor out of three different T cell donors.

**Figure 5 cells-09-01264-f005:**
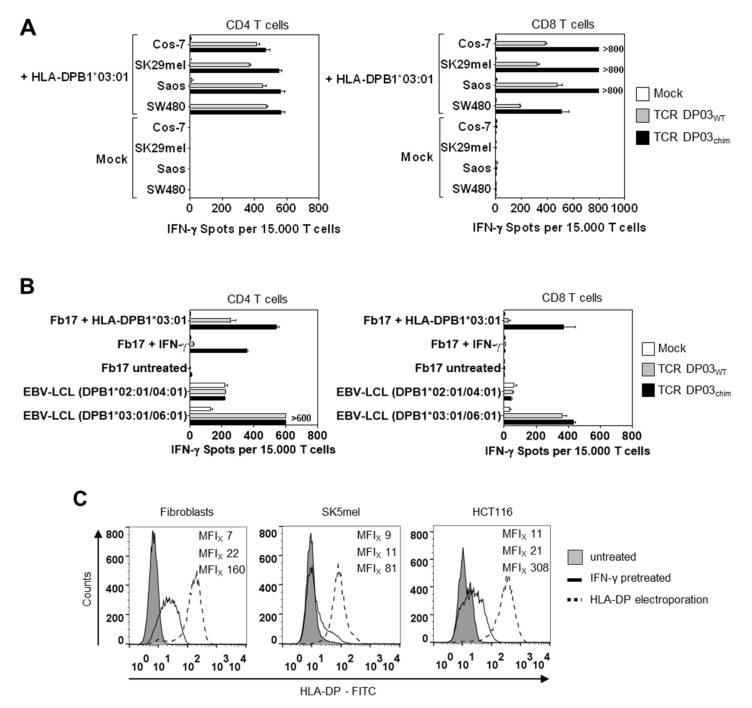
Impact of HLA-DP cell surface expression level on T cell recognition. (**A**) Cos-7, SK29mel, Saos, and SW480 tumor cell lines were transfected with HLA-DPA1*01:03 and -DPB1*03:01 IVT-RNA and used as target cells in IFN-γ ELISpot assay. Pre-stimulated CD4 (left panel) and CD8 (right panel) T cells from an HLA-DPB1*03:01 negative donor were transfected with TCR DP03_WT_, TCR DP03_chim_ or Mock and incubated with target cells in an E: T ratio of 0.3:1. (**B**) IFN-γ spot formation by indicated T cell populations upon incubation with HLA-DPB1*03:01 positive primary fibroblasts (Fb) from donor 17 that were either pretreated with IFN-γ or electroporated with HLA-DPA1*01:03 and -DPB1*03:01 coding RNA. EBV-LCL from HLA-DPB1*03:01 positive and negative donors served as controls. (**C**) HLA-DP surface expression on HLA-DPB1*03:01 positive cells (primary fibroblasts, SK5mel, and HCT116) was determined by flow cytometry. Cells were left untreated (filled grey line), pretreated with IFN-γ for four days (solid black line) or electroporated with RNA encoding HLA-DPA1*01:03 and -DPB1*03:01 (black dotted line). MFI values of HLA-DP staining are shown in the following order: untreated (upper value), IFN-γ pretreated (middle value), HLA-DP electroporated (lower value). (**D**) Primary AML samples from individual HLA-DPB1*03:01 positive patients were analyzed cytofluorometrically for HLA-DP expression. MFI values of HLA-DP staining were shown for unstained (upper value) or HLA-DP stained (lower value) cells. (**E**) IFN-γ spots formed upon stimulation of indicated CD4 and CD8 T cell populations by tumor cell lines SK5mel and HCT116 that were either left untreated, IFN-γ pretreated or electroporated with HLA-DPA1*01:03 and -DPB1*03:01 encoding RNA at an E:T ratio of 0.6:1. In addition, untreated primary AML blasts from individual HLA-DPB1*03:01 positive and negative patients were also used as target cells. Standard deviations of means of IFN-γ spot formation are shown for two technical replicates and experiments were performed with two (**A**) or three (**B**,**E**) different T cell donors, from which one representative is depicted.

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
