# Peer review of "HLA-DPB1 Reactive T Cell Receptors for Adoptive Immunotherapy in Allogeneic Stem Cell Transplantation"

_cells, 2020, doi:10.3390/cells9051264_

Round 1

Reviewer 1 Report

This is an interesting study about the possible role of modified T cells expressing anti-DP TCR for anti-cancer responses.  While it might be expected that CD4+ cells were the cells mediating anti-MHC class II directed lysis, no indication is given as to the cytolytic pathway used by the CD4+ cells to mediate lysis.  Is it the granule exocytosis pathway or is it via expression of FasL or TRAIL?

Reviewer 2 Report

Klobuch et al report work describing the use of HLA-DPB1 specific TCRs in adoptive immunotherapy in allogenic stem cell transplantation. They used patient wild type and genetically optimized TCRs and performed in vitro and some mouse in vivo work.

Experiments included the use of T cells with two main TCRs (DP04 in figures 1-3 and DP03 in figures 4-5). Their characterization of these T cells (cytokine production, cytotoxicity, anti-leukemic control) provides findings on potential uses of these TCRs and also issues with cross reactivity of INF treated fibroblasts that must be dealt with from a safety standpoint. The manuscript is well written and provides important findings but has some deficiencies to address.

1) There are multiple places in the manuscript where “data not shown” is used (including lines 166, 183, 186, 202 241, 263, and 290). These should all be supplemental figures since they are experiments or important controls necessary for evaluating the manuscript findings and for other researchers doing similar experiments.

2) Figure 2 should be divided into an A and B to make it easier to follow what is being referred to in the results. The CD8 data should be included as supplemental figure.

3) With the issue of cross reactivity to pre-treated fibroblasts, it would be useful to include whether or not there were any signs of self-reactivity detected in the mouse study described in figure 2.

4) K562 cells are discussed, but the type of cell that they are should be included in the manuscript along with why they are the cell line of choice for these experiments.

5) Figure 3 can be labeled better. It takes a while to understand that figure 3A is only CD4 cells so that can be labeled better. Why is the CD8 cell data against these HLA-DP transfected K562 cells not shown?

6) TCR DP04 is characterized in figures 1-3 (IFN-g, tumor lysis, anti-leukemic control in AML engrafted mouse model, and reactivity towards different HLA-DP alleles on leukemic and healthy cell lines). The researchers moved onto the characterization of TCR DP03 in a similar way in figures 4-5 because it appeared to be less cross reactive, but it is missing the mouse experiments. Addition of this data would strengthen the manuscript.

7) The discussion would be improved by adding additional perspective on how the avidity was believed to be increased in these studies. Also, expanding the discussion of the role of avidity in cross reactivity and removal of the target cells would strengthen the paper.

Reviewer 3 Report

In the manuscript entitled “HLA-DPB1 reactive T cell receptors for adoptive immunotherapy in allogeneic stem cell transplantation” by Klobuch et al., the authors describe the isolation of HLA-DPB1 specific T cell receptors and determine their activity in transduced CD4 and CD8 T cells towards primary AML blasts by using IFN-γ ELISpot as well as cytolytic activity assays and murine AML xenograft models. The isolated wildtype TCR as well as an optimized murinized TCR with higher expression levels showed strong AML reactivity in vitro and, at least when transducing CD4 T cells, were able to effectively eliminate leukemia blasts in an AML xenograft NSG mouse model proving the therapeutic potential of such an approach. However, the optimized TCR and under some conditions the wildtype TCR were also reactive to non-hematopoietic cells indicating off-target effects of such engineered T cells.

While a role for such engineered T cells is getting more and more established for adoptive immunotherapy, the work does provide some additional and important information for this field. The study is well designed with appropriate description of the methods used. The experiments are well performed with proper controls and results are presented in a clear manner. However, as a minor point, some results, which are mentioned in the manuscript but just referred to as “data not shown”, should be included as supplementary figures. This is specifically recommended for data showing HLA-DP expression in K562 cells (mentioned on page 6, lines 202 and 203) as well as SK29mel, Saos, SW480 and Cos-7 cells (page 9, lines 288,289) after transfection. In addition, flow cytometry plots of TCR expression levels of CD4 and CD8 T cells transfected with the murinized TCR DP03chim should be shown (page 8, line 260).

If those minor concerns are addressed, I would definitely recommend the manuscript to be published in Cells.

Round 2

Reviewer 2 Report

Thanks for making the requested changes.